# Gonad Ontogeny and Sex Differentiation in a Poeciliid, *Gambusia holbrooki*: Transition from a Bi- to a Mono-Lobed Organ

**DOI:** 10.3390/biology12050731

**Published:** 2023-05-16

**Authors:** Komeil Razmi, Ngoc Kim Tran, Jawahar G. Patil

**Affiliations:** 1Laboratory of Molecular Biology, Fisheries and Aquaculture Centre, Institute for Marine and Antarctic Studies, University of Tasmania, Taroona, TAS 7053, Australia; komeil.razmi@utas.edu.au (K.R.); ngockim.tran@utas.edu.au (N.K.T.); 2Department of Aquaculture, Faculty of Agriculture and Natural Resources, An Giang University, a Vietnam National University Ho Chi Minh City, Long Xuyen City 880000, Vietnam

**Keywords:** sexual identity, pest fish, gonadogenesis, germ cells, gonadosoma

## Abstract

**Simple Summary:**

This study maps critical events of gonadogenesis in the live-bearing fish *Gambusia holbrooki*. Significantly, it provides the first evidence for primary gonochorism, the transition of an embryonically bi-lobed gonad to a mono-lobed organ, as occurs in adults and the dimorphically programmed onset of sex differentiation in this species. The outcomes are significant in basic biology and are applicable for developing control options for this notorious pest fish.

**Abstract:**

Despite their uniqueness, the ontogeny and differentiation of the single-lobed gonads in the poeciliids are very poorly understood. To address this, we employed both cellular and molecular approaches to systematically map the development of the testes and ovary in *Gambusia holbrooki* from pre-parturition to adulthood, encompassing well over 19 developmental stages. The results show that putative gonads form prior to the completion of somitogenesis in this species, a comparatively early occurrence among teleosts. Remarkably, the species recapitulates the typical bi-lobed origin of the gonads during early development that later undergoes steric metamorphosis to form a single-lobed organ. Thereafter, the germ cells undergo mitotic proliferation in a sex-dependent manner before the acquisition of the sexual phenotype. The differentiation of the ovary preceded that of the testes, which occurred before parturition, where the genetic females developed meiotic primary oocytes stage I, indicating ovarian differentiation. However, genetic males showed gonial stem cells in nests with slow mitotic proliferation at the same developmental stage. Indeed, the first signs of male differentiation were obvious only post-parturition. The expression pattern of the gonadosoma markers *foxl2*, *cyp19a1a*, *amh* and *dmrt1* in pre- and post-natal developmental stages were consistent with morphological changes in early gonad; they were activated during embryogenesis, followed by the onset of gonad formation, and a sex-dimorphic expression pattern concurrent with sex differentiation of the ovary (*foxl2*, *cyp19a1a*) and testes (*amh* and *dmrt1*). In conclusion, this study documents for the first time the underlying events of gonad formation in *G. holbrooki* and shows that this occurs relatively earlier than those previously described for ovi- and viviparous fish species, which may contribute to its reproductive and invasive prowess.

## 1. Introduction

In vertebrates, gonadogenesis typically commences at late segmentation/early pharyngula stage, marked by homing of PGCs at the gonadal ridge [1,2,3,4]. Mechanistically, this is orchestrated by a combination of chemotaxis cues [5,6,7], topology of the somatic environment [8,9] and tuning of cell–cell adhesion [10,11], resulting in PGC colonisation and their compaction at the gonad primordia. Following this, gonadosoma cells emerge at the presumptive gonad, skirt the colonised PGCs and promote sexual differentiation in sync with the genotype of the germline. Within this conserved gonadogenesis plan, teleosts have evolved varying sex differentiation strategies [12] governed by intrinsic, e.g., sex chromosomes [13] and PGC population machinery [14], and/or are influenced by extrinsic factors, e.g., environmental [15] and social [16]. Despite the diversity in sex determination mechanisms (e.g., XX/XY or ZZ/WZ), the sex differentiation machinery maintains the sexual identity of gonads through a cascade of downstream sex-regulating genes, which are highly conserved between vertebrates [17,18].

Among teleosts, poeciliids stand out with their unique patterns of gonadogenesis and reproductive strategies such as internal fertilization [19], single-lobed gonad formation [20], production of spermatozeugmata [21,22], prolonged ovarian storage of spermatozoa [23] and superfetation [24]. Nonetheless, the mechanisms governing these reproductive adaptations in poeciliids are poorly understood; however, they are of significant interest from both basic and applied points of view, particularly in the eastern mosquitofish, *Gambusia holbrooki*. *G. holbrooki* is a small but notorious freshwater pest fish, which is listed by IUCN among the top 100 alien pest species that has been spreading within and across invaded aquatic habitats [25]. With internal fertilisation and superfetation, gambusia embryos have enhanced survival rates due to combined placental (matrotrophy) and yolk (lecithotrophy) nourishment, which allow them to parturiate as fully developed free-swimming larvae [26,27]. Although the growth rate and maturity in *G. holbrooki* are correlated with temperature, salinity and food availability [28,29], in optimum conditions, they are primed for puberty as early as six-week post-fertilisation [26,30], contributing to their reproductive and invasive prowess. Hence, gaining insight into the cellular and genetic pathways of their germ cell development and gonadogenesis is expected to enhance our understanding of their reproductive biology. Particularly, the recent discovery of a sex marker in this species [18,31] allows the identification of sexual genotypes of the embryos and hence investigations into early events of sex differentiation, i.e., far before the phenotypic sex is discernible. Such knowledge is expected to facilitate genetic control of *G. holbrooki* pest populations [32], including scheduling and fine-tuning hormone treatment regimens for the manipulation of sex as well as providing new insights into other reproductive vulnerabilities. 

Thus, as a first step towards discerning the early events of gonadogenesis in *G. holbrooki*, this study employed morphological and molecular surrogates. In particular, critical events in germ cell homing, the morphological transformation of early gonads, the onset of meiosis and sex-specific patterns of gene expression were investigated in genetically male and female *G. holbrooki*. 

## 2. Materials and Methods

### 2.1. Fish Collection and Processing

Wild-caught Gambusia were transported from the Tamar Island Wetland Reserve (41°23.1′ S; 147°4.4′ E) to the Institute for Marine and Antarctic Studies (IMAS), Taroona, University of Tasmania, Australia. The developing embryos were obtained from newly caught gravid females and staged as previously described [19]. To obtain larvae and juveniles, females with known gravid spot intensities [24] were individually allocated to a static tank set up with a breeding trap to protect neonates from cannibalisation. After parturition, the females were moved to stock tanks, each clutch of juveniles was raised in a 2 L static tank (21 ± 0.5 °C, 16 L: 8 D-lights turned on at 06.00 h) and periodically sampled until 150 days post-parturition (dpp). Newborn fish were only fed with freshly hatched Artemia nauplii (INVE Aquaculture, Salt Lake City, UT, USA) for 30 dpp and later supplemented with commercial micro-pellets (Nutra XP 0.5, Skretting, TAS, Australia) until the end of the experiment. Based on our earlier work [4,19,31], nineteen developmental stages/time points were used to document critical stages of gonad development and sex differentiation: gastrula, early segmentation, late segmentation, pharyngula, parturition, 1, 2, 5, 10, 15, 20, 25, 30, 50, 70, 90, 110, 120 and 150 dpp.

### 2.2. Embryonic Staging and Gonad Histology

The embryos were harvested surgically from wild-caught gravid females and staged as recently described for this species [19]. The anatomical morphology of germ cells and gonad were studied by staining tissue sections with hematoxylin and eosin (H&E) according to standard protocols [33]. Briefly, the samples were fixed in Bouin’s solution (Sigma-Aldrich, Sydney, NSW, Australia) overnight and rinsed in 70% ethanol for several hours. Tissue processing, including dehydration, clearing and paraffin infiltration, was conducted by a tissue processor Tissue-Tek^®^ vacuum infiltration processor (Miles Laboratories, Pittsburgh, PA, USA). The 3 µm sections of the specimen were deparaffinised in xylene, dehydrated in progressive concentrations of ethanol, stained by H&E and mounted using Pertex^®^ (Histolab, Gothenburg, Sweden). The sections were imaged under a Leica DM750 (Wetzlar, Germany) light microscope and processed using the Leica Application Suite software (version 3.8.0, Wetzlar, Germany). The number of PGCs were counted in both sexes from corresponding developmental stages for comparison. Germ cell counts were carried out using the cell counter plugin ImageJ [34] on serial transverse and sagittal sections (*n* = 6/each axis) for a given sample and the numbers computed by multiplying the average cell numbers in transverse and the sagittal sections. 

### 2.3. Genetic Sexing of Embryos and Neonates

The genetic sex of the embryos was identified by adopting a simplex PCR assay described previously for the species [18]. Briefly, the genomic DNA of embryos was obtained from tail fin clips (pharyngula, parturating embryos and larvae) or paraffin-embedded specimens (gastrula and segmentation embryos) using the QIAamp DNA FFPE Tissue Kit (QIAGEN, Valencia, CA, USA). The genetic sex of embryos was determined by polymerase chain reaction (PCR) using sex-specific genetic markers as described [18,31]. Briefly, the PCR reaction (15 μL) comprised 1 × MyTaq™ HS Red mix (Meridian Life Science, Memphis, TN, USA), 1.5 μM of each primer and 50 ng of genomic DNA template. Thermal cycling (T100™ Thermal Cycler, Bio-Rad Laboratories, Inc., Sydney, NSW, Australia) consisted of 95 °C for 1 min, followed by 30 cycles of 95 °C for 5 s, 60 °C for 5 s and 72 °C for 20 s. Sex-dimorphic amplicons were visualised and separated using gel electrophoresis. 

### 2.4. Cloning and Characterisation of Key Gonadosoma Markers

The expression pattern of four genes involved in gonadosoma function, *dmrt1*, *amh*, *cyp19a1a* and *foxl2*, were studied at five distinctive time points of development, including late segmentation, pharyngula, parturition and at 12 and 30 days post-parturition (dpp). The full-length cDNA of three genes with known gonad function, i.e., *dmrt1*, *foxl2* and *cyp19a1a*, along with housekeeping genes, i.e., *pgk1*, *rps18* and *beta-actin*, were cloned using the RACE technique as previously described [4]. Briefly, all publicly available genetic sequences for the genes of the respective live-bearing poeciliids, namely *Poecilia latipinna*, *P. mexicana*, *P. formosa*, *P. reticulata* and *Xyphophorus maculatus*, were aligned using MUSCLE [35] and two pairs of degenerate primers designed to clone partial CDSs using nested PCR to obtain species-specific partial gene sequences. In parallel, total RNA, isolated from gonads, were used to generate GeneRacer^TM^ cDNA libraries (Invitrogen, Carlsbad, CA, USA). Subsequently, a pair each of primary and nested primers for amplification of 5′ and 3′ regions were designed based on the partial cDNA sequence and used to amplify the cDNA ends, whose identity was verified by sequencing and BLAST [36] homology.

### 2.5. RNA Extraction, Reverse Transcription and Quantitative PCR (qPCR)

Developing embryos and larvae were dissected from brooding mothers and individually fixed in RNA*later*™ (Invitrogen, Melbourne, VIC, Australia) overnight. For RNA isolation, the embryos were individually homogenised with 19 and 27 G needle-syringe aspirations and the total RNA isolated using Isolate II RNA Micro Kit (Bioline, Narellan, NSW, Australia). The gDNA contamination was eliminated from RNA samples using Ambion DNaseI (Invitrogen, Waltham, MA, USA) and the DNase-treated samples were purified by a column-based method. RNA integrity was qualified by visualising the integrity of the 18S and 28S ribosomal RNA on a 1% agarose gel stained with SYBER^TM^ Safe (Invitrogen, Waltham, MA, USA). Total RNA concentration and genomic DNA contamination were measured using the Qubit™ 4 Fluorometer with RNA HS and DNA BR Assay Kits (Invitrogen, Waltham, MA, USA), respectively. Approximately 1 µg of total RNA was used for reverse transcription using the Tetro^TM^ cDNA Synthesis Kit (Bioline, Narellan, NSW, Australia). The 20 µL reaction contained 200 units of reverse transcriptase, 0.5 mM dNTP mix, 10 units of RNase inhibitor, 0.5 mM oligo (dT)_18_ primer mix and 1X reverse transcriptase buffer. The samples were incubated at 45 °C for 60 min and the enzyme was deactivated at 85 °C for 5 min.

The qPCR assays were designed to amplify 85–130 bp amplicons (Table 1). The qPCR mix comprised 1X iTaq Universal SYBR Green Supermix (Bio-Rad, Sydney, NSW, Australia), 5–10 ng cDNA Template and 0.4 µM of each primer and was adjusted to 20 µL using MilliQ water. Triplicate reactions were run for each cDNA sample using the CFX96 Touch Real-Time PCR Detection System (Bio-Rad, Sydney, NSW, Australia) and consisted of 95 °C for 1 min, followed by 40 cycles of denaturation at 95 °C for 5 s, annealing (see Table 1 for temperatures) for 15 s, and extension at 72 °C for 7–15 s. For positive and negative controls, 5 ng cDNA (from a tissue where the target gene is expressed) and MilliQ water were used as templates, respectively. Amplicon identities were confirmed by melt curve analysis, gel visualisation and sequencing. 

### 2.6. qPCR Data Normalisation and Statistical Analysis

For biological normalisation of data from the qPCR assay, the stability of *rps18*, *gapdh*, β-*actin* and *pgk1* [37,38] was first tested by geometric averaging [39] using the geNorm algorithm [39,40] in qbase^+^ software, version 3.0 (Biogazelle, Zwijnaarde, Belgium). As a result, *gapdh* was selected as the most stable (M value = 0.45) housekeeping gene that is highly expressed but not sexually differentiated. The relative transcription of target genes was calculated using the comparative threshold cycle (Cq) method with efficiency correction [41]. The relative expression of genes of interest (∆Cq) was calculated against the selected reference gene (*gapdh*). The fold changes were measured using the 2^−∆∆Cq^ method and presented in box plots. 

A one-way analysis of variance (ANOVA) was used to compare means within and between developmental stages and sexes. The Tukey post hoc test was applied wherever population means were significantly different. The level (*p* < 0.05, *p* < 0.01, or *p* < 0.001) of significance between two means was also determined. All statistical analyses were performed and plots generated using OriginPro, Version 2020 (OriginLab Corp., Northampton, MA, USA). 

### 2.7. Whole-Mount In Situ Hybridisation (WM-ISH)

The ontogeny of germ cells and the formation of gonads were examined by detection of the *vasa* transcript using whole-mount chromogenic in situ hybridisation (WM-CISH) in developing embryos [4]. Briefly, the digoxigenin-labelled antisense *vasa* RNA probe was generated through in vitro transcription from an approximately 1.15 kb coding region using the T3/T7 RNA polymerase (NEB, Sydney, NSW, Australia) and DIG RNA labelling mix (Roche, Mannheim, Germany). The cDNA template and unincorporated reaction components were removed from the RNA probe using Ambion DNase I (Invitrogen, Waltham, MA, USA) and column purification, respectively. The ready probe was stored with 0.5 U/µL of RNasein^®^ Plus Ribonuclease Inhibitor (Promega, Fitchburg, WI, USA) at −20 °C until use. The late segmentation, pharyngula and parturating embryos were dissected from gravid females, rinsed with cold PBS and fixed using 4% paraformaldehyde (Emgrid, Adelaide, SA, Australia) overnight at 4 °C. The fixed embryos were rinsed in PBT (PBS containing 0.1% Tween 20) and dehydrated with progressive concentrations of methanol replaced with PBT. Subsequently, the specimens were permeabilised with 20 µg/mL proteinase K (EO0491, Thermo Scientific, Waltham, MA, USA) for 15–25 min at 37 °C and post-fixed with 4% paraformaldehyde for 20 min at room temperature. The embryos were prehybridised at 68 °C for 3 h in hybridisation buffer (50% formamide, 5X SSC, 0.01% Tween 20, Torula Yeast tRNA, 50 µg/mL heparin). The samples were incubated in a hybridisation buffer containing 1 ng/µL antisense *vasa* probe at 68 °C for 16 h. Non-specifically bound and excessive probes were removed through consecutive desalting stringency washes using PBST. For DIG labelling, first, the non-specific regions for anti-DIG antibodies were blocked using a blocking solution comprising a 5% blocking reagent (11096176001, Roche, Basel, Switzerland) dissolved in 1X maleic acid buffer containing 0.1% Tween 20 (MABT) for several hours. The embryos were then incubated in a 1:3000 dilution of Anti-Digoxigenin-AP (alkaline phosphatase), Fab fragments (11093274910, Roche, Basel, Switzerland) in a blocking solution overnight at 4 °C. The antibody-labelled embryos were washed with PBT 8 times for 30 min each (with gentle agitation), washed in staining buffer (100 mM Tris HCl pH 9.5, 50 mM MgCl_2_, 100 mM NaCl, 0.1% Tween 20) for 30 min and incubated in BM-purple (Roche). The chromogenic AP substrate was then added to develop dark purple hybridisation signals to the desired contrast. The embryos were then washed in PBT until all excess stain was removed. Finally, the embryos were postfixed with 4% paraformaldehyde overnight at 4 °C and washed and stored in PBS for imaging by light microscopy using the Leica DM750 (Wetzlar, Germany) and imaging software Leica Application Suite, version 3.8.0 (Wetzlar, Germany). 

## 3. Results

### 3.1. PGC Colonisation Occurs before Complete Somitogenesis

The migrating PGCs were first identified at early to mid somitogenesis (*n* = 4 embryos), concurrent with the first visible tailbud, 15 somites, optic bud formation and segregation of the central nervous system into telencephalon, diencephalon and mesencephalon. At this stage, the migrating PGCs were localised ventral to the pronephric cells and adjacent to the developing intestine (Figure 1A,B).

The migrating PGCs were also noticeably larger (diameter 13.4 ± 0.5 µm; *n* = 31) than other surrounding cells, and they were characterised by barely stained cytoplasm and a conspicuously distinguishable nucleus that occupied most of the cell volume. The first observation of colonised PGCs (Figure 1C,D) was captured in late somitogenesis (*n* = 17 embryos) that coincided with the onset of melanophore pigmentation, otolith enlargement, tail movement and detached eye cups from the yolk. At late somitogenesis, the gonad primordia formed a clutch of populating germ cells attached to the coelomic epithelial wall via connective tissue that extended ventrally and posteriorly (Figure 1C,D). The newly colonised PGCs retained their morphological characters; however, they were stained lighter (Figure 1B″,C″,D′) compared with migrating PGCs (Figure 1A″). Nevertheless, the staining capacity of homed germ cells was regained when they underwent mitosis at the pharyngula stage (see Figure 2E). The morphology of PGCs, the timing of their colonisation and primitive gonad emergence (*n* = 11, F:M = 7:4) did not show any sex-dimorphic pattern at late somitogenesis.

### 3.2. Mitotic Proliferation of Germ Cells

At the onset of the pharyngula stage—where melanophores on the dorsal fin, pulsating mouth and blood circulation throughout internal organs were observed—the germ cells had colonised the distinct putative gonad lobes (Figure 2A,A′). The germ cells also underwent passive mitotic proliferation, and their abundance was comparable between male (Figure 2B, *n* = 9) and female (Figure 2C, *n* = 6) embryos (*p* > 0.05). In addition, somatic cells were infrequently observed throughout the presumptive gonad of both genetic males (Figure 2B) and females (Figure 2C). At this developmental stage, germ cell proliferation in both sexes was marked by type I mitosis, i.e., characterized by only individual germ cells (Gc I) skirted by somatic cells (Figure 2B,C). Here, the number of homed germ cells in females at the genital ridge was not significantly different (mean = 200 ± 18, *n* = 7, *p* > 0.05; Figure 2F) compared to males (mean = 118 ± 5, *n* = 9; Figure 2F). The earliest significant difference (*p* < 0.05) in germ cell numbers between genetic males (mean = 355 ± 37, *n* = 7; Figure 2F) and females (mean = 1148 ± 56, *n* = 6; Figure 2F) was first observed at mid-pharyngula, concurrent with jaw enlargement, the emergence of caudal fin ray elements and teeth on the mandible. Here, in males, undifferentiated (presumed spermatogonial) stem cells underwent type I mitotic proliferation (Figure 2D) and resulted in a significant increase in germ cell numbers in comparison with early pharyngula (Figure 2F, *p* < 0.05). At mid pharyngula stage, apart from type I, type II (cyst-forming) mitotic oogonia were also identified in genetic females (*n* = 5, Figure 2E). The type II mitosis was evident by synchronously developing nest of cells containing two or four daughter cells with interconnected cytoplasm. At this stage, the somatic cells also populated the putative ovary and appeared to skirt germ cell nests.

### 3.3. Differentiation of Gonads

Just before parturition, while the gonad lobes were in the closest vicinity to each other (Figure 3A,A′), meiosis was first observed in genetically female embryos (*n* = 17) by the presence of stage I primary oocytes arrested at prophase I (Figure 3B). This coincided with the developing caudal fin reaching the forebrain and the formation of anus and urogenital pores, with the concurrent appearance of five teeth in the premaxilla. At this stage, some embryos (*n* = 8) also had a few primary oocytes at stage II, spread sporadically throughout the ovary and surrounded by a single layer of follicle cells. Mitotic oogonia were also found in the meiotic ovary, mostly located in the periphery of the tissue (Figure 3B′). At the same developmental stage, gonad development in genetically male embryos (*n* = 9) appeared accelerated through stem cell (presumed spermatogonia) proliferation that formed clusters of gonial cells with a homogenous morphology (Figure 3C). The emerging Sertoli cells were first observed in the presumptive testis of these genetic males as triangle-shaped cells (Figure 3C′) that were nested between putative spermatogonial cells. However, neither the testicular germ cells nor their soma showed any of the morphological characteristics of sex differentiation until 5 dpp. By 10 dpp, spermatogonia had appeared in clusters of more than 4 cells enclosed by precursors of Sertoli cells (Figure 4A–A″). In addition, stromal cell aggregations, which would later develop into sperm ducts and testicular interstitial compartments, were first seen in the gonads at this stage. By 15 dpp, spermatogonia appeared in clusters of more than 8–16 cells in a clone (Figure 4B) and a slit-like opening was found inside the central region of the stromal cell aggregations (Figure 4B″). By 20 dpp, the male gonads mostly contained spermatogonia clusters. At this stage, meiosis/spermatogenesis had also commenced; several cysts of primary spermatocytes were observed, and sperm ducts were formed (Figure 5A–A″).

### 3.4. Post-Natal Gonad Development

In genetic females, the juvenile gonad (i.e., 1 to 20 dpp) remained at its pre-natal status, containing primary oocytes in the chromatin–nucleolus stage as was already evident from mid-pharyngula (Figure 5B). However, the elongation of somatic tissue that initiated the formation of the ovarian cavity (Figure 5B′) was first evident in the 15 dpp gonads.

From 20 to 120 dpp, in male gonads, testicular lobules and the entire testis increased in length and width due to the formation of a larger number of cysts. At 20 dpp, the male gonads contained mainly cysts of spermatogonia (Figure 5A–A″), while at 120 dpp, the testis was mainly occupied by primary spermatocytes (Figure 6A,A′). In female gonads, from parturition embryos to 120 dpp, oocytes in the chromatin–nucleolus stage continued to grow by increasing the amount of cytoplasm and the size of nucleus (Figure 6B). At this stage, oogenesis was arrested in the diplotene stage of prophase I, with primary oocytes characterised by a round-oval nucleus and surrounding follicle cells (Figure 6B′). 

At 150 dpp, males had developed a fully functional gonopodium with a well-developed elongated testis (Figure 6C,C′), significantly larger than those of the 120 dpp males. The elongation of testicular lobules was due to the increasing number of cysts that contained spermatocytes and spermatids. At this stage, the spermatozoa packets (spermatozeugmata) were observed for the first time, accumulated in the lumen of the sperm duct. At 150 dpp, ovaries slightly increased in size compared to those of 120 dpp females, due to the transition of primary oocytes to previtellogenic oocytes, with abundant oil droplets in their ooplasm (Figure 6D,D′). In *G. holbrooki*, females have a cyst ovarian type, i.e., a single and saccular ovary forms with a central lumen and well-developed cavities which appear in the dorsal side of the ovaries within 4 months of post-natal development.

### 3.5. Gross Morphological Fusion of the Gonad Lobes

Embryonic gonadogenesis in *G. holbrooki* underwent a significant morphological transformation through laterally converging axial lobes, ensuing in a single-lobed organ at post-parturition. As can be seen from *vasa*-positive cells, germ cells began to coalesce at two distinct ridges after colonisation at late somitogenesis stages (Figure 7A). Later, at the pharyngula stage, the two primary lobes grew convergently and were enlarged by populating germ cells; however, they were distinctly separate from each other until the late pharyngula stages (Figure 7B). Just before parturition, the primary lobes were observed in the closest vicinity to each other, with the right lobe appearing smaller than the left (Figure 7C), and they eventually merged into a single-lobed gonad in juvenile males (Figure 5A and Figure 6A) as well as juvenile females (Figure 5B, Figure 6C and Figure 7D). The pattern of gonad transformation, including the orientation and fusion of primary lobes (right to left), remained unchanged among different individuals and genders (*n* = 17, F:M = 7:6).

At the mid-pharyngula stage (n = 5 embryos, F:M = 4:1, Figure 8A), both presumptive gonad lobes could be seen in the single sagittal plane but are interconnected with the gonadal stroma (Figure 8A′). The merging lobes appeared connected to the coelomic wall and spleen through connective tissue. These germ-cell islands later fused through gonadal stroma expansion and germ-cell proliferation in parturating embryos, developing into a single gonad lobe in both females and males.

The morphological transformation of the male and female gonad structures is summarised in Figure 9. At late segmentation, the paired testes hung from the basal part of mesentery by the mesorchium (Figure 9A), but then the testes drew very close together just after birth (Figure 9B) and, at around 5 dpp, finally fused into a singular lobe at the hilar region before germ cells of the right testis migrated to the left testis and proliferated to form a triangular prism-shaped gonad (Figure 9C). By 10 dpp, as the testis was growing, two clusters of stromal cells aggregated in the centre of the testis (Figure 9D). These stromal cell aggregations then formed two testis ducts medially at around 20 dpp (Figure 9E). By 150 dpp, the testis obtained its adult size, where a network of efferent ducts was surrounded by secretory testicular tissue located predominantly on the right side of the abdomen next to the intestine (Figure 9F). In females, at late segmentation, the paired lobes of the ovary was suspended from the basal part of the mesentery by the mesovarium (Figure 9G), but these ovaries then drew closer together progressively (Figure 9H). By 5 dpp, the lobes fused into a lobe at the hilar region before the germ cells of the right lobe migrated to the left and proliferated to form a triangular prism-shaped ovary (Figure 9I). By 15 dpp, a slit appeared at the dorsal side of the fused ovary, surrounded by stromal cells (Figure 9J). By 30 dpp, the lateral side of the dorsal stroma elongated upward along the coelomic wall to form an ovarian cavity (Figure 9K). The formation of the ovarian cavity began from the anterior region of the ovary and proceeded caudally. After completing the fusion of the paired ovaries, an oviduct formed around 70 dpp (not shown in Figure 9) as the fused ovary elongated in a caudal direction. By 150 dpp, there was a W-like shaped ovarian cavity presenting at the dorsal side of the ovary (Figure 9L).

### 3.6. Quantitative Expression Patterns of Gonadosoma Markers

At late somitogenesis, concurrent with PGC homing, *foxl2* expression was not detected (Figure 10A), while *cyp19a1a*, *amh* and *dmrt1* were detectable (Figure 10B–D). However, the patterns were comparable between genetic males and females (Figure 10E). At the pharyngula stage, concurrent with the mitotic proliferation of germ cells, the expression patterns of *dmrt1*, *amh* and *cyp19a1a* were not sex-dimorphic. Although the *foxl2* expression was detected for the first time at the pharyngula stage in both sexes; it was significantly higher in females (∆∆Cq = 4.8 ± 0.81 log_2_, *p* < 0.01; Figure 10A). Right before parturition, coinciding with sex differentiation in genetic females (i.e., the onset of meiosis in primary oocytes), *cyp19a1a* expression was significantly higher in females (∆∆Cq = 5.6 ± 0.52 log_2_, *p* < 0.001; Figure 10A) compared with the males. In the same developmental stage, *dmrt1* expression in genetic females underwent a significant suppression (∆∆Cq = 3.7 ± 0.55 log_2_, *p* < 0.05; Figure 10D) to more than 12-fold lower compared to genetic males. Nevertheless, *amh* did not show a noticeable sex-dimorphic expression at the same stage (Figure 10C).

Similar to pre-natal stages (Figure 10E), the relative expression of *foxl2* in the juvenile (∆∆Cq = 5.1 ± 0.52 log_2_, *p* < 0.001; Figure 10B) and the adult ovaries (∆∆Cq = 4.5 ± 0.75 log_2_, *p* < 0.001; Figure 10B) was much higher than those of post-natal testes. The expression of *cyp19a1a* was significantly higher in the post-natal ovary, displaying an increasing trend between the juvenile (∆∆Cq = 3.2 ± 0.44 log_2_, *p* < 0.05; Figure 10A) and adult phases (∆∆Cq = 4.2 ± 0.76 log_2_, *p* < 0.01, Figure 10A). Interestingly, the relative expression of *cyp19a1a* in the post-natal ovary was not as abundant as at the parturition stage. The relative expression of *amh* at the post-natal phase was significantly male-biased; however, its expression was most skewed in the juvenile phase, showing the highest sex-dimorphic difference (∆∆Cq = 5.2 ± 0.62 log_2_, *p* < 0.05; Figure 10C) within the studied window. Similarly, *dmrt1* expression maintained a male-biased pattern at the post-natal phase, with noticeable downregulation in the juvenile (∆∆Cq = 6.0 ± 0.57 log_2_, *p* < 0.01, Figure 10D) and adult ovaries (∆∆Cq = 6.1 ± 0.62 log_2_, *p* < 0.01, Figure 10D).

## 4. Discussion

This study has documented the critical time-points of gonad ontogenesis, its morphological transformation and sex differentiation for the first time in a live-bearing teleost. In parallel, the activation and expression pattern of key gonadosoma markers validated these reproductive developmental events, providing unique comparative insights between teleosts. Specifically, a developmental framework between PGC colonisation and gonadosoma apparition immediately before gonial mitosis was established, including accurate quantification of the newly homed PGCs.

### 4.1. Morphology and Clustering of Germ Cells Is Sex-Dimorphic in Undifferentiated Gonads

The number of colonised PGCs was not sex-biased in the early gonad of *G. holbrooki*, similar to those of medaka, *Oryzias latipes* [42], zebrafish, *Danio rerio* [43] and the western mosquitofish, *Gambusia affinis* [20]. However, the clustering pattern of germ cells is sex-dimorphic post-colonisation, where undifferentiated gonads undergo mitosis. The observation of both type I and II mitotic proliferation of germ cells in embryonic gonads of *G. holbrooki* was comparable to those reported for medaka [44], catfish, *Tachysurus ussuriensis* [45], rainbow trout, *Oncorhynchus mykiss* [46] and zebrafish [47], suggesting these early events are largely conserved across teleosts. Specifically, the type I mitosis of germ cells resembled stem cell-like self-renewal division as described in medaka [48]. Later, at mid-pharyngula, the retention of a self-renewal division of germ cells in the presumptive testis (i.e., in genetic males) was similar to that observed in other teleosts [17,49,50]. Similarly, as observed in other teleosts, the presumptive ovary (i.e., genetic females) underwent type II mitosis (i.e., gametogenesis-committed division), which is the earliest sign of ovarian differentiation [51], although no primary oocyte was yet observed. This onset of type II mitosis can be attributed to the relatively higher number of germ cells observed in genetic females than males. A similar increase in germ cell numbers of developing ovaries has been reported in medaka [44,52], three-spined stickleback, *Gasterosteus aculeatus* [53] and zebrafish [43].

### 4.2. Ovarian Differentiation Precedes Testis Differentiation and Occurs Earlier than in Most Teleosts

A clear indication of ovarian differentiation, i.e., the occurrence of meiotic oocytes, in *G. holbrooki* was observed immediately before parturition, unlike the presumptive testis, which only had stem cell-like gonial clusters. This demonstrates that sex differentiation in females occurs much earlier than males in this species. In teleosts, the onset of ovary differentiation varies between species. For example, in egg layers such as the Japanese medaka, primary oocytes are first observed immediately after hatching [54]. In *O. mykiss*, sex differentiation is observed 16–29 days post-hatching, immediately after complete yolk sac absorption and the beginning of oral feeding [55]. In *Cyprinus carpio*, this is postponed to 70 days post-hatching, concurrent with the onset of the juvenile stage [56]. In the live-bearing species *G. affinis,* ovarian differentiation occurs two days before parturition [20], similar to *G. holbrooki*. Even in some poeciliids such as *Zoarces viviparus* [57] and *P. reticulata* [58], ovary differentiation is delayed up to 12–18 days after parturition. The relative early gonad differentiation in *G. holbrooki* and its sister species *G. affinis* may explain their high reproductive capacity and, consequently, their well-known invasive propensity [25]. 

As demonstrated, *G. holbrooki* is a primary gonochorist, where the testis and ovary show the first sign of differentiation as early as mid-pharyngula. Contrastingly, its closely related species, including *G. affinis* [20] and *P. reticulata* [59,60], exhibit secondary gonochorism, i.e., all offspring initially differentiate as females before acquisition of their sexual fate. This supports the observation that even the evolutionarily related species exhibit different sex differentiation mechanisms [61,62] that are also later mirrored in their reproductive strategies. 

### 4.3. Gonadosoma Markers Forecast the Timing of Sex Differentiation

The female-biased expression of *foxl2* at the onset of meiosis in genetic females of *G. holbrooki* suggests a role in ovarian differentiation, as also occurs in the rainbow trout *Oncorhynchus mykiss* [63]. A similar role for *foxl2* has been attributed in medaka, although its expression is restricted to females [64]. More importantly, *foxl2* mutation causes female-to-male sex reversal in the Nile tilapia *Oreochromis niloticus* [65], confirming that it is required for ovarian differentiation. However, similar perturbation experiments will be required to validate its role in *G. holbrooki*. The peak expression of *cyp19a1a* in female embryos of *G. holbrooki* also at the onset of meiosis indicates a role in sex differentiation, similar to what has been shown in medaka [66] and zebrafish [67]. Similarly, the strong expression of *dmrt1* and *amh* coinciding with the onset of meiosis, an event that occurs later (at the post-natal stage) in males, is in agreement with delayed testis differentiation in this species. This supports the previous observation that the *G. holbrooki* testis retains its sexual plasticity post-parturition [31]. 

The expression of *dmrt1* in the pre- and post-natal ovary of *G. holbrooki*, albeit at low levels, is similar to what occurs in zebrafish [68]. This suggests that *dmrt1* may not be a sex-determining gene, but its sex-dimorphic expression serves as an indicator for early detection of sex in this species. Similarly, in zebrafish, the expression level of *dmrt1* in bi-potential gonads is an early marker for the prediction of sexual phenotype [69]. However, the autosomal *dmrt1* expression in medaka ovary begins 20 days post-fertilisation [70] and is restricted to mitotic oogonia [71]. In addition, a *dmrt1* mutation in medaka causes the development of a functional ovary in XY individuals, suggesting that it plays a critical role in testis differentiation [72]. 

The co-expression of *foxl2* and *cyp19a1a* in the ovary of *G. holbrooki* is conserved among fish species and supports their mutual role in ovarian differentiation. For instance, in medaka, *foxl2* is co-expressed with *cyp19a1a* in the surrounding germ cells as early as in ten days post-hatching [64]. Indeed, in *O. niloticus*, *foxl2* is known to function upstream of ovarian aromatase by enhancing the activity of Ad4BP/SF, a transcription factor which regulates *cytochrome P450* genes [73,74]. The diametrically opposite and sex-dimorphic expression of *foxl2* (high in females and low in males) and *dmrt1* (high in males and low in females) in *G. holbrooki* gonads suggests an antagonistic relationship between the two genes. This is in agreement with the roles that *foxl2* [75] and *dmrt1* [76] play in the differentiation and maintenance of the ovary and testis, respectively. For example, repressing *Foxl2* in adult mice results in the expression of *Dmrt1* in granulosa cells, accompanied by the appearance of testicular tissue, i.e., structures resembling seminiferous tubules in the ovary [77]. Conversely, repression of *dmrt1* results in differentiation of male (Sertoli cell) to female (granulosa cell) gonadosoma in human testes [78].

In practice, information on the temporal expression of the genes involved in sex differentiation could assist in fine-tuning the window of exogenous hormone administration to produce mono-sex populations of *G. holbrooki*. For example, the onset of *foxl2* expression at pharyngula and *cyp19a1a* later at parturition defines a narrow window before which the embryos are susceptible to masculinisation. This requires the treatment of babies via the brooding mothers (Patil, personal observations). In contrast, the genes involved in male differentiation, i.e., *dmrt1* and *amh*, are at their peak later between parturition and the juvenile stages, suggesting that feminising hormone treatment can be delayed until post-parturition. In support of this, feminisation in this species has been achieved by treating the embryos post-parturition for 30 days [25], an aspect that can be fine-tuned, based on the temporal expression patterns of *dmrt1* and *amh*. Hence, an effective masculinisation treatment must be administered via the mothers, i.e., before female embryos have undergone sex differentiation in *G. holbrooki*. However, in poeciliid species such as *P. sphenops* [79], post-natal treatment of testosterone is sufficient for effective masculinisation, suggesting species-specific differences.

### 4.4. Embryonic Recapitulation of the Bi-Lobed Gonad

Typically, the gonads in teleosts are bi-lobed, except for poeciliids [80]. As observed in *G. holbrooki*, the gonads initially develop as a bi-lobed organ; however, the lobes are fused later at the post-natal stage to form an integrated organ. This occurs at around the same time in both males and females. However, in some other poeciliids, the lobes are not completely fused, even in adulthood. For example, in *P. Mexicana*, the testes remain as paired lobes in adults, although the lobes are in the closest vicinity to an incomplete fusion [81]. Regardless, this morphological fusion of gonad lobes appears to have evolved for supporting viviparity in poeciliids, i.e., integrating two lobes to form an anterior ovarian section to produce gametes and a dorsal and posteriorly extending uterus-like cavity for gestation. However, the fusion of the testes is intriguing, and it implies that the structural scaffold of early gonadogenesis is laid far in advance of sex differentiation. In practice, this may explain the sexual plasticity and susceptibility/feasibility of the species to hormonal sex reversal. It must be noted that the morphological details of gonad formation are not completely conserved among poeciliids. Instead, they show varying degrees of gonad fusion [81], implying their comparative utility in deciphering the evolutionary links that lead to the more specialised ovary and uterus of higher vertebrates. 

## 5. Conclusions

This study shows for the first time that sex differentiation in *G. holbrooki* follows that of a primary gonochorist, where the ovary and testis initially differentiate into distinct organs. This was supported by both the cellular events as well as the expression pattern of key molecular surrogates. The relatively early occurrence of critical gonadogenesis events such as PGC colonisation (at the late segmentation stage), the sex-dimorphic expression pattern of gonadosoma markers, acquisition of sexual phenotype and ovarian maturation in *G. holbrooki* reflect its rapid reproductive capability which may contribute to the invasive capacity [82] of this species. Unlike other teleosts, which maintain their fertility throughout adult life, the poeciliids undergo reproductive senescence [83,84], similar to menopause in mammals. The early sex differentiation (prior to parturition) observed in *G. holbrooki* may compensate for fertility loss later in life, accounting for its invasive capacity. However, the molecular mechanisms of their reproduction and invasive capacity remain poorly understood. Future studies encompassing perturbation experiments may provide insights into genetic and reproductive susceptibilities that may be exploited to suppress their reproductive output and hence their invasion. Importantly, as poeciliids form a key intermediate node between higher vertebrates and teleosts, the outcomes of this study enhance our comparative knowledge of vertebrate reproductive development and its evolution. 

## Figures and Tables

**Figure 1 biology-12-00731-f001:**
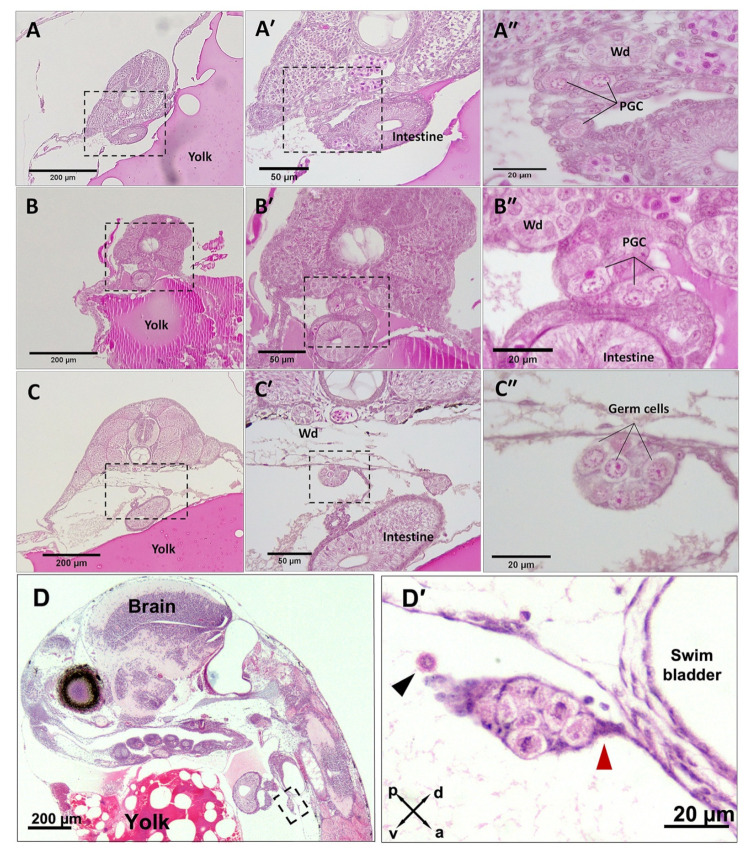
PGC migration and colonisation in *G. holbrooki*. The cross sections of embryos at mid segmentation (**A**–**A″**,**B**–**B″**) show migrating PGCs adjacent to the Wolffian duct (Wd) and early intestine. Sagittal sections (**C**–**C″**,**D**–**D′**) show subsequent migration of PGCs to the abdominal cavity and colonisation of the genital ridge, at late segmentation. The colonised PGCs were attached to the coelomic wall by connective tissue ((**D′**), red arrowhead) located ventral to the swim bladder (**D′**). A migrating PGC ((**D′**), black arrowhead) at the abdominal cavity is in the process of merging with the homed germ cell precursors. Panels (**A**–**C″**)—dorsal to the top and ventral to the bottom. Crossed double-headed arrows in (**D′**) show the relative orientation: a, anterior; p, posterior; d, dorsal and v, ventral.

**Figure 2 biology-12-00731-f002:**
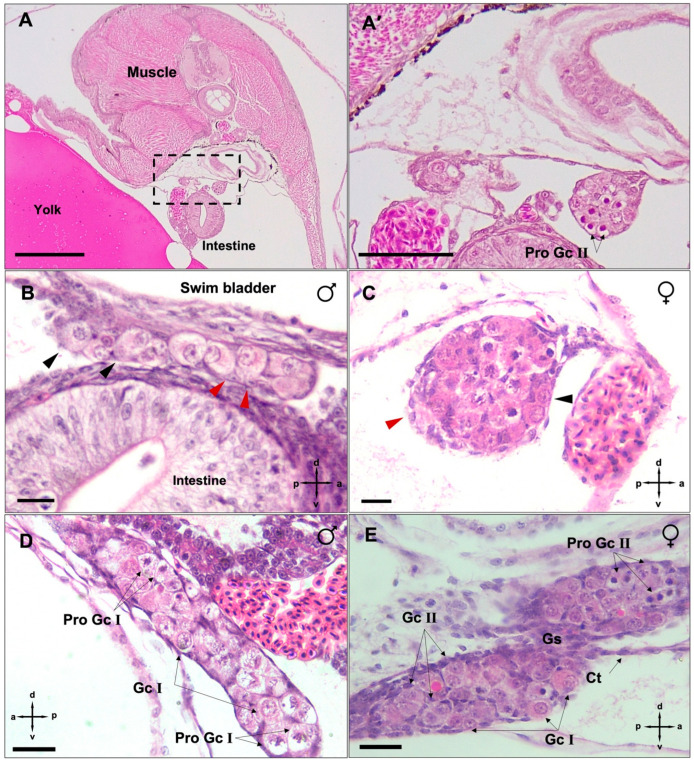
Panels showing the sex-dimorphic pattern of germ cell proliferation in *G. holbrooki*. At early pharyngula, the female gonad lobes formed (**A**,**A′**). However, testicular germ cells remained in their precursor form ((**B**), black arrowhead), maintaining a relatively large size and low staining capacity. Somatic cells ((**B**,**C**), red arrowhead) were also observed at the putative gonads of both sexes. In early pharyngula females (**C**), proliferating germ cells were characterised by a highly stained nucleus. The individual germ cells, skirted by somatic cells ((**C**), black arrowhead) is indicative of type I mitosis. At mid-pharyngula, the genetic males showed stochastic mitotic proliferation type I and slight tissue enlargement (**D**). In genetic female embryos (**E**), oogonia nests with two daughter cells interconnected through the cytoplasm and surrounded by a layer of somatic cells are indicative of type II mitotic proliferation (Pro Gc II). The number of colonised PGCs at late segmentation was comparable between the sexes but differentiated at the pharyngula stage (**F**). The asterisks indicate the level of significance between groups (** *p* < 0.01, *** *p* < 0.001). Gc I, germ cell type I; Gc II, germ cell type II; Pro Gc, proliferating germ cells; Gs, gonadal stroma; Ct, connective tissue; NS, not significant. Scale bars = 20 µm.

**Figure 3 biology-12-00731-f003:**
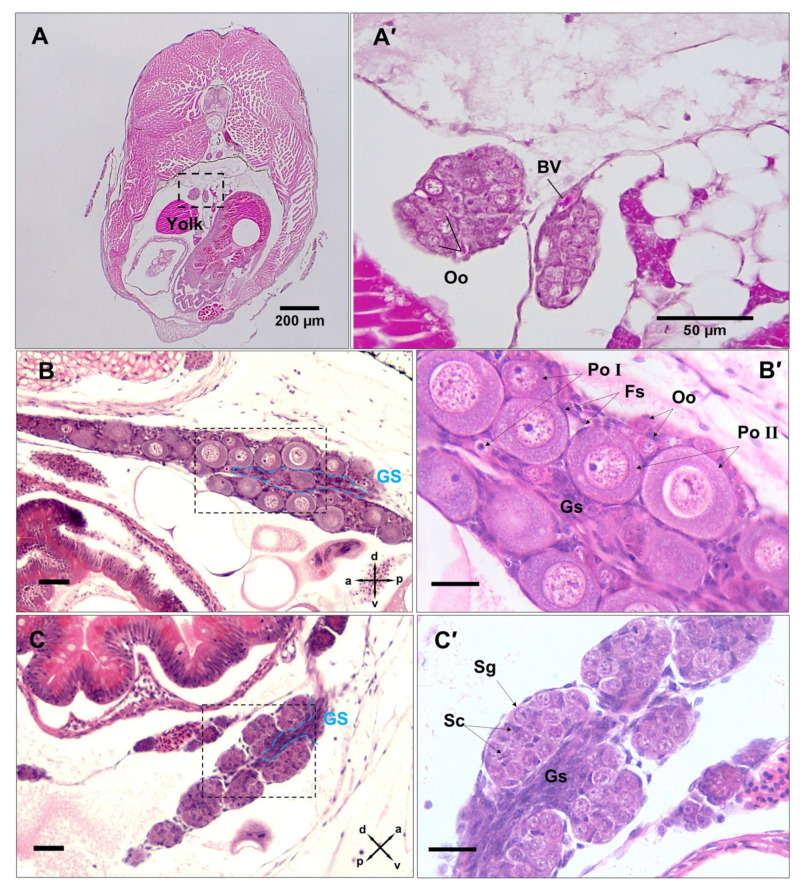
Histomicrographs showing earliest signs of sex differentiation in female *G. holbrooki*. The undifferentiated ovary was bi-lobed post pharyngula (**A**,**A′**), with differentiation occurring immediately before parturition with nearly fused lobes (**B**,**B′**), characterised by emerging primary oocytes stage I. Some female embryos also had primary oocyte stage II at the same developmental stage (**B′**). Before parturition, the ovary was dominated by meiotic germ cells, with the occasional presence of mitotic oogonia at the periphery. The somatic follicle cells of the ovary formed as a single layer surrounding primary oocytes. The pre-parturition gonad in genetic males was fused, however, it remained undifferentiated (**C**,**C′**) and was characterised by nests of mitotic germline stem cells throughout the gonad, with the first occurrence of the precursor of Sertoli cells nested between spermatogonial cells (**C′**). Abbreviations: oogonia, Oo; blood vessels, BV; primary oocyte stage I, Po I; primary oocyte stage II, Po II; gonadal stroma, Gs; Sertoli cells, Sc; spermatogonia, Sg; Follicle cells, Fs. Scale bars = 25 µm.

**Figure 4 biology-12-00731-f004:**
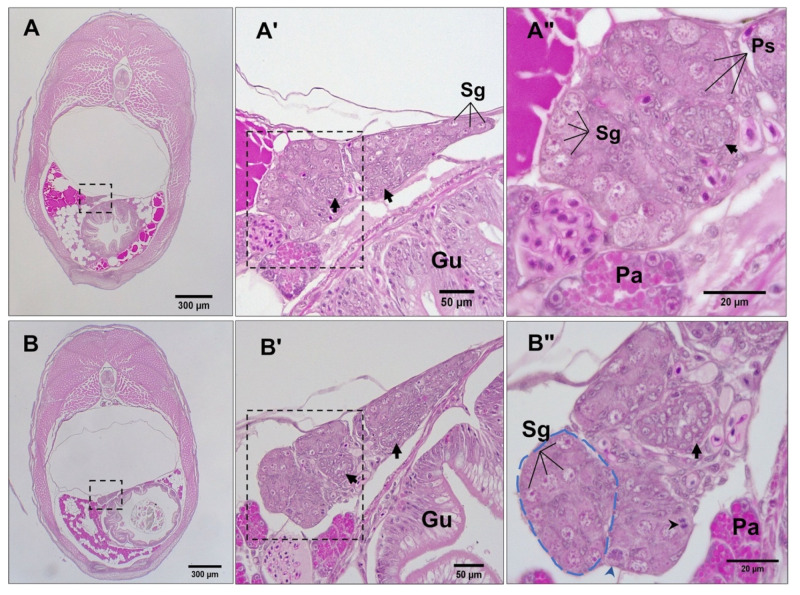
Histomicrographs of transverse sections (caudal to cranial view) through 10 (**A**–**A″**) and 15 (**B**–**B″**) dpp male *G. holbrooki*. The gonads (see broken-line frames in (**A**,**B**)) were suspended in the coelom and located laterally to the left of the gut. (**A′**,**A″**) are magnified views of areas framed in (**A**,**A′**), respectively, showing aggregations of stromal cells (black arrows) and the presence of spermatogonia (Sg) enclosed by pre-Sertoli cells (Ps). (**B′**,**B″**) are magnified views of areas framed in (**B**,**B′**) respectively, showing the formation of sperm ducts (black arrows) and acinar structures (blue dashed line) containing spermatogonia in nests, wrapped by pre-Sertoli cells. Spermatogonia at mitotic prophase (blue arrowhead) and at anaphase (black arrowhead) can also be seen. Gu, gut; Pa, pancreas.

**Figure 5 biology-12-00731-f005:**
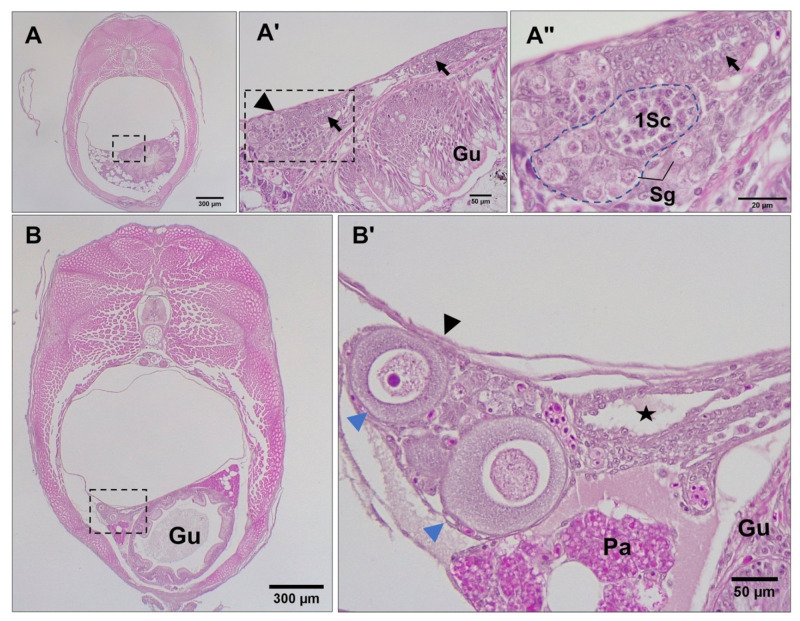
Histomicrograph showing abdominal transverse sections (caudal to cranial) through 20 dpp males (**A**–**A″**) and females (**B**,**B′**). The gonads are located dorsolateral to the gut and dorsally tethered to the mesovarium ((**A′**,**B′**), black arrow heads). The testis shows the sperm ducts (black arrows) and the formation of testicular lobes ((**A″**), framed by blue dashed line) that contained acinar structures formed by nests of spermatogonia at the blind termini of the lobes, and primary spermatocyte cysts. In the ovary, primary oocytes in the chromatin–nucleolus stage (blue arrowheads) were found (**B′**). There was shaping of the ovarian cavity ((**B′**), black star) towards the right. Sg, spermatogonia; 1Sc, primary spermatocytes; Pa, pancreas; Gu, Gut.

**Figure 6 biology-12-00731-f006:**
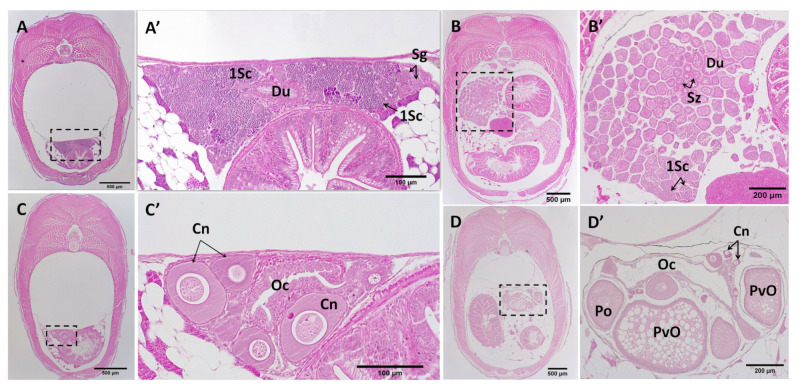
Histomicrograph showing abdominal transverse sections (caudal to cranial) through 120 (**A**,**C**) and 150 (**B**,**D**) dpp, males (**A**,**B**) and females (**C**,**D**,**D′**). In the juvenile testis (**A**,**A′**), the spermatogonia were located at the periphery and were dominated by spermatocytes. The juvenile ovary (**C**,**C′**) was mainly occupied by primary oocytes arrested at prophase I. In the adult testis (**B**,**B′**), spermatogenesis occurred at the periphery with migration of maturing germ cells centrally towards the efferent duct. Spermatozoa packaged in bundles (spermatozeugmata) were released into the testis ducts (Du). In the adult ovary (**D**,**D′**), oocytes in multiple stages of development are marked. Du, sperm duct; 1SC, primary spermatocytes; Sg, spermatogonia; Sz, spermatozeugmata; Oc, ovarian cavity; Po, primary oocyte; PvO, previtellogenic oocyte.

**Figure 7 biology-12-00731-f007:**
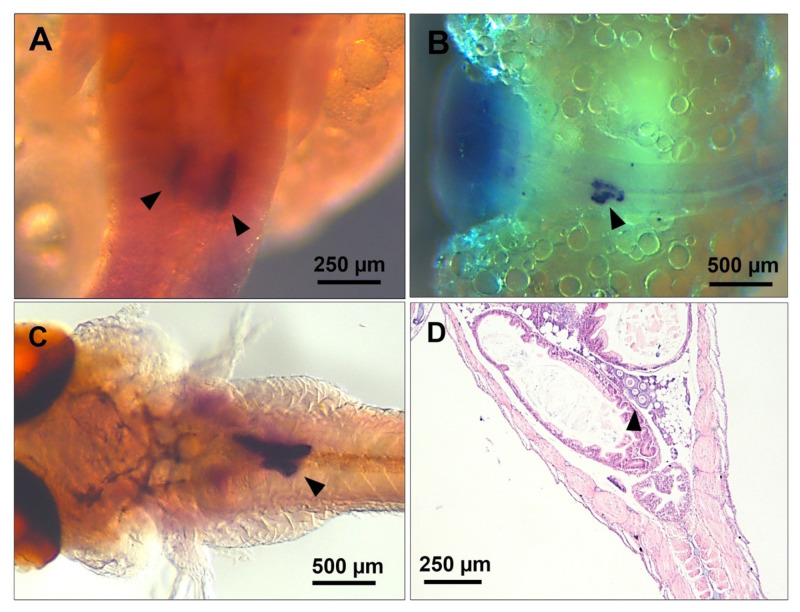
ISH Micrographs showing axial transformation and fusion of presumptive gonad lobes during late embryogenesis. The *vasa*-labelled germ cells (black arrowheads) first colonised two distinct ridges on either side of the body axis during late somitogenesis (**A**). Subsequently, the lobes grew larger and closer at mid-pharyngula (**B**), with the right lobe diminishing in size, indicative of its merging with the left, just before parturition (**C**). The H&E-stained micrograph (**D**) shows a mono-lobular gonad (black arrowhead) in a 12 dpp juvenile female. (**A**,**C**), ventral views; (**B**,**D**) dorsal views.

**Figure 8 biology-12-00731-f008:**
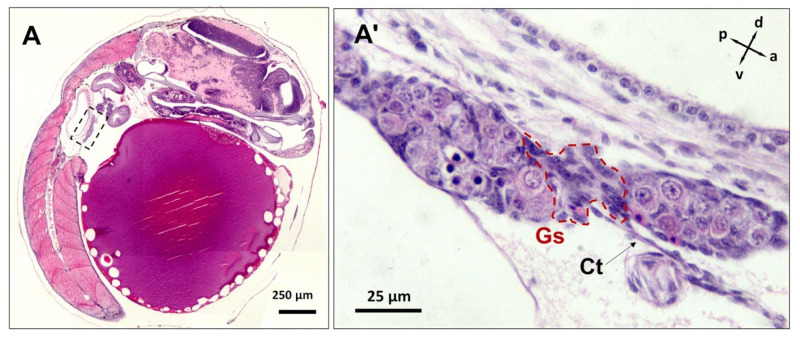
The H&E-stained micrographs of developing embryos at the mid-pharyngula stage (sagittal sections). The location of the putative gonads is framed ((**A**), dashed box) and magnified (**A′**). The formation of two germ-cell islands in the same plain is indicative of the two merging lobes that are interconnected through gonadal stroma (Gs; framed by red dotted line) and attached to the coelomic wall via connective tissue (Ct).

**Figure 9 biology-12-00731-f009:**
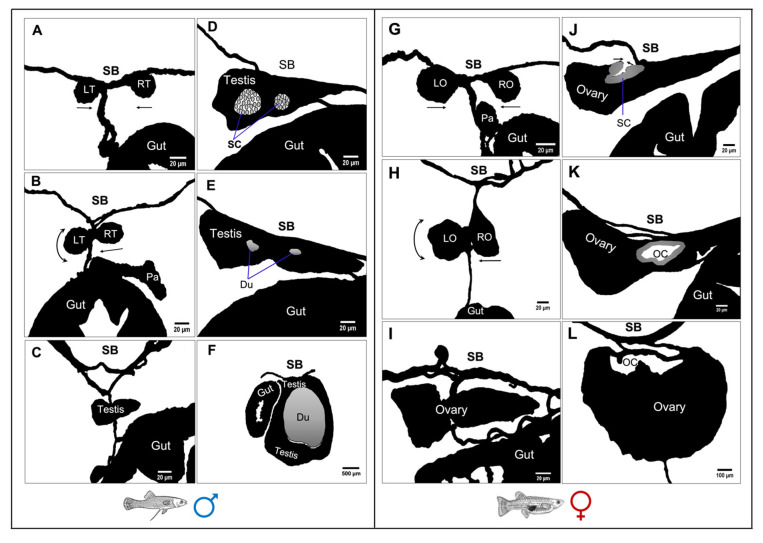
Schematic diagrams showing the formation of a single-lobed testis (left panel) and ovary (right panel) in *G. holbrooki* (caudal to cranial view). At the late-segmentation stage, paired testes (**A**) and ovaries (**G**) were suspended from the dorsal coelomic wall by the mesorchium and mesovarium, respectively. Progressively (leading to the parturition stage), the separate testis (**B**) and ovarian (**H**) lobes were drawn very close to each other, with the right lobe tending to merge with the left and with the left lobe shifting anteriorly. By 5 dpp, the two lobes were fused at the hilar region, with germ cells migrating from the right to the left lobe and proliferating to form a triangular cross-section (**C**,**I**). By 15 dpp, the lateral side of the dorsal stroma had elongated upward along the coelomic wall in the testis (**D**) and the ovary (**J**). At around 30 dpp, elongated lateral sides of dorsal stroma fused to form two efferent ducts (**E**) and a single cavity (**K**) in the ovary. By 150 dpp, the testis (**F**) with central efferent ducts was located beside the gut, while a well-developed ovarian cavity was observed at the dorsal part of the ovary (**L**). Du: efferent ducts, G: gut, O: ovary, OC: ovarian cavity, P: pancreas, SB: swim bladder, SC: somatic cell cluster, LR: left testis, RT: right testis, LO: left ovary, RO: right ovary.

**Figure 10 biology-12-00731-f010:**
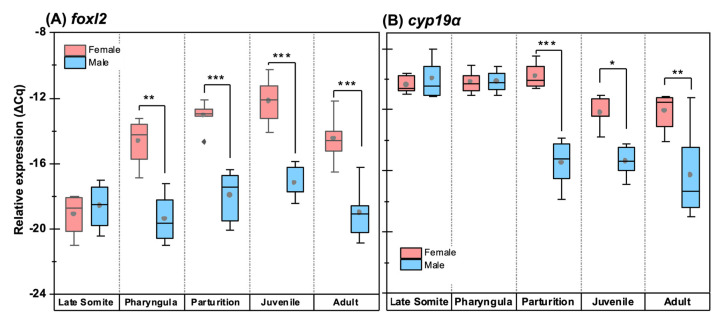
Relative (normalised) expression of four genes (*dmrt1*, *amh*, *foxl2* and *cyp19a1a*) involved in sex differentiation at five developmental stages. Boxplots (**A**–**D**) display the relative expression (∆Cq) of the respective genes in males and females. The asterisk indicates the level of significance between groups (* *p* < 0.05, ** *p* < 0.01, *** *p* < 0.001). The heatmap (**E**) shows the sex-dimorphic fold change of gene expression at the corresponding developmental stages.

**Table 1 biology-12-00731-t001:** Primer sequence, amplicon size, qPCR annealing temperature and source of primers used for qPCR.

Gene		Primer Sequence 5′–3′ *	Amplicon Size (bp)	Annealing Temperature (°C)	Reference/ Accession ID
House keeping	*beta-actin*	F—CGGCAGGACTTCACCTACAGACACCT	99	68	[4]
		R—CTTGCACAAACCGGAGCCGTTGTCA			
	*gapdh*	F—AGCCAAGGCTGTTGGCAAGGTCATC	133	67.5	[18]
		R—GTCATCATACTTGGCTGGTTTCTCC			
	*pgk1*	F—GATGATCATCGGTGGCGGCATGG	96	66.5	OL988673
		R—ATACAGCGCCTTCCTCGTCGAACA			
	*rps18*	F—GGAGAGGCTGAAGAAGATCAGGGCTC	109	66.5	OL988674
		ACCGACAGTGCGACCACGACG			
Male biased	*dmrt1*	F—CACCCTTCGTCAGCCTGGAGGAGA	85	67.0	OL988671
		R—ATGGTCGAGTCGTAGCTGGTAGGTGAA			
	*amh*	F—CCCCTGCAGATGGAGAGCTGGGCGTCATTT	88	64	[18]
		AACGTCGTCCCTGAARTGCAAGCAGA			
Female biased	*cyp19a1a*	F—GCTTGTGGAGGAGATGAGCACGGTT	97	63.5	(Patil, personal collection)
		R—CATCACTTTCAGTCTTTCATAACTGACG			
	*foxl2*	F—GCAAAGGGAGAGGCAGAGGAGGA	108	66.0	OL988672
		R—CTCTACCGCCTCTCCCACTGAAACCA			

* F, Forward primer; R, Reverse primer.

## Data Availability

All relevant data is included in the manuscript.

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
