# Peer review of "Gonad Ontogeny and Sex Differentiation in a Poeciliid, Gambusia holbrooki: Transition from a Bi- to a Mono-Lobed Organ"

_biology, 2023, doi:10.3390/biology12050731_

Round 1

Reviewer 1 Report

Comments and Suggestions for Authors

The authors systematically mapped the development of both testes and ovary in Gambusia holbrooki, from pre-parturition to adulthood, and analyzed the expression pattern of the gonadosoma markers, foxl2, cyp19a1a, amh and dmrt1, in pre- and post-natal developmental stages. Some results are interesting and novel. However, some statements of detailed analyses and figure description could be improved to make all manuscript more sense.

Comments:

1. Line 94-97: ‘Nineteen developmental stages/time points were used to document spatio-temporal events of gonad development and sex differentiation’, why the author chose these 19 stages instead of other stages?

2. In Table1, some genes such as gapdh, amh and cyp19a1a, the Tm of these primers was larger difference between F and R, so how to perform qPCR? The qPCR method needs to be described.

3. Figure 1, the authors showed partial image of the embryos, a complete embryo image need be provided to better help readers understand the position of PGCs.

4. Line 261: How to identify type I and II germ cells, its characteristics need to be described in detail.

5. Figure 3, what did Cn, BV and Sg mean?

6. Line 313, "However, the elongation of somatic tissue that initiated the formation of ovarian cavities were first evident in the 15 dpp gonads (Fig. 5Bʹ)." But in figure 5, "Histo-micrograph showing abdominal transverse sections (caudal to cranial) through 20 dpp males (A-Aʺ) and females (B and Bʹ)." So, its 20 dpp or 15 dpp?

7. Figure 4, the authors just showed the male sections of 10 and 15 dpp, need to supplement female sections.

8. In figure 7D, "The H&E stained micrograph (D) shows a mono-lobular gonad (red arrowhead) in a 12 dpp juvenile female." I did not find the red arrowhead, just a black arrowhead. And the number of primary oocytes in a 12 dpp juvenile female (figure 7D) were more than that in a 20 dpp juvenile female (figure 5B'), why?

9. All of the image resolution needs to be improved.

Author Response

Please see attached file,  which addresses the comments/suggestions of all the three reviewers

Reviewer 2 Report

This manuscript systematically studied the critical events of gonadogenesis and the expression pattern of sex-determining genes in the live-bearing fish Gambusia holbrooki, and got rich and high quality results, especially provided the import evidence that the transition of an embryonically bi-lobed gonad to a mono-lobed organ as occurs in adults.

However, two points as followed need to answer and provide more information.

1. There were 19 developmental stages of fish to be studied, but the results just emphasized on the PGC colonisation of 4 developmental stages. If too many results in the manuscript, it is better to show the tissue histology of all investigated stages via the attached files. It is also easier for the reader to understand the basis for choosing the four important stages.

2. In the part of 3.5 on line 350, “which eventually merged into a single-lobed gonad in juvenile fish (Fig. 7D) in both sexes.” But in Fig.7D, it was only showing the H&E stained micrograph in a 12 dpp juvenile female.

Author Response

Please see attached, which addresses the comments/suggestions of all three reviewers

Reviewer 3 Report

This paper is about the development of 18 both testes and ovary in Gambusia holbrooki, from pre-parturition to adulthood. My comments as follow:

Abstract: well elaborate

Introduction: accepted

Materials and methods: has scientific merits

Result: present clearly

Discussion: accepted

Conclusion: please provide research gap and future work

References: Please make sure 80% of references are within 5 years

Need to have minor revision

Author Response

Please see attached, which addresses comments/suggestions of the three reviewers

Round 2

Reviewer 1 Report

The author has already responded to most of the questions, but there are two important issues that need to be explained, otherwise they cannot be accepted.

1. In Table1, the author modified the TM of all primers, but the primer sequence has not changed. The author must provide sufficient explanation.

2. Figure 4, the authors just showed the male sections of 10 and 15 dpp, need to supplement female sections.

Author Response

Comment 1 : In Table1, the author modified the TM of all primers, but the primer sequence has not changed. The author must provide sufficient explanation.

Response: We clarify that the Tm were not modified. Instead, the column was changed to Annealing temperature, in response to your earlier comment. Please note that the column heading now reads as Annealing Temperature NOT Tm. We believe this is self-explanatory. No changes made.

2. Figure 4, the authors just showed the male sections of 10 and 15 dpp, need to supplement the female sections.

Response: We are unsure what the issue is. As responded before, this figure aims to show the changes in male fish, where significant changes occurred i .e.,  showing differentiation. As responded before, the ovary remained remarkably similar to that observed at 1 DPP (i.e. already differentiated) and there is little value in presenting this figure. We make no changes.